# Comparison between Conventional and Digital Workflow in Implant Prosthetic Rehabilitation: A Randomized Controlled Trial

**DOI:** 10.3390/jfb15060149

**Published:** 2024-05-31

**Authors:** Massimo Corsalini, Giuseppe Barile, Francesco Ranieri, Edvige Morea, Tommaso Corsalini, Saverio Capodiferro, Rosario Roberto Palumbo

**Affiliations:** 1Department of Interdisciplinary Medicine, ‘Aldo Moro’, University of Bari, 70100 Bari, Italy; massimo.corsalini@uniba.it (M.C.); saverio.capodiferro@uniba.it (S.C.); 2Department of Prosthodontics, Magna Graecia Institute, 74121 Taranto, Italy; francescoranieri9@gmail.com (F.R.); dr.robertopalumbo@gmail.com (R.R.P.); 3Department of Medicine and Aging Science, University ‘G. D’annunzio’, 66100 Chieti, Italy; vivimorea@gmail.com

**Keywords:** implant prosthesis, digital dentistry, digital workflow

## Abstract

The progress of digital technologies in dental prosthodontics is fast and increasingly accurate, allowing practitioners to simplify their daily work. These technologies aim to substitute conventional techniques progressively, but their real efficiency and predictability are still under debate. Many systematic reviews emphasize the lack of clinical RCTs that compare digital and traditional workflow. To address this evidence, we conducted a three-arm designed clinical RCT, which compares fully digital, combined digital, and analogic and fully analog workflows. We aimed to compare the clinical properties of each workflow regarding interproximal (IC) and occlusal contact (OC), marginal fit, impression time (IT), and patient satisfaction through a VAS scale. In total, 72 patients were included in the study. The IC and OC of the digital workflow were better than the others (*p* < 0.001), which obtained similar results. No difference between implant–abutment fit was observed (*p* = 0.5966). The IT was shorter in the digital workflow than the others (*p* < 0.001), which were similar. Patient satisfaction was higher in the digital workflow than in the conventional one. Despite the limitations, this study’s results support better accuracy and patient tolerance of digital workflow than of conventional techniques, suggesting it as a viable alternative to the latter when performed by clinicians experienced in digital dentistry.

## 1. Introduction

Biomechanical engineering and computer science development have enabled the introduction of three-dimensional images in both the design of surgical plans and dental treatments. Digital dentistry is constantly evolving and capable of changing therapeutic strategies, clinical methods, and materials that can be used in everyday practice. The scientific literature contains several studies supporting digital dentistry in daily clinical practice, hypothesizing its wide diffusion in the years to come [1,2,3]. The advantages are considerable, in terms of execution, patient comfort, and accuracy, but their comparison with analogic flow is still under debate [4]. Much progress has occurred in digital technologies to address the disadvantages, aiming at a progressive substitution of conventional and analog techniques. In particular, implant-supported crowns can nowadays follow an entire digital flow, from surgical planning to prosthesis finalization. This process consists of an initial digital impression with an intraoral scanner (IOS), combined with Cone Beam Computed Tomography (CBCT) to plan the surgical implant insertion through dedicated software. After that, CAD-CAM technology is helpful to create a surgical guide to allow the correct planned implant placement. Then, a digital impression is performed with scan abutment, and the definitive prosthesis is milled or printed after a CAD design, giving the clinician a reliable alternative to conventional procedures.

Nevertheless, some analogic steps are still required, often resulting in a combined digital–analogic workflow to achieve the most predictable result for the patient. The traditional workflow may include digital planning, but the impression, the model casting, and the prosthesis manufacturing are performed with analogic materials, such as elastomers for the impression and analog procedures for layering the ceramics onto plaster models. Analysis of the precision and accuracy of the digital, mixed, and conventional methods is widespread in current literature, but there is an important lack of clinical studies because most of the studies are in vitro [5]. Moreover, the results of clinical studies are inconsistent. For single implant-supported crowns, authors report a better accuracy of the entire digital workflow [6,7,8]. Conversely, no significant differences have been found between digital and conventional workflows [9,10,11]. On the other hand, the systematic reviews conducted on this topic agree on the time efficiency of digital impressions, which are faster and better tolerated by patients [1,5,12,13]. Therefore, no general consensus exists, and we recognize the need to address this lack of knowledge with an RCT to deepen the clinical comparison between digital and analogical workflows.

This study aims to perform a three-arm randomized controlled trial (RCT) comparing entirely digital, combined digital–analogic, and entirely analogic workflows of implant-supported prostheses through the evaluation of interproximal (IC) and occlusal contact (OC), marginal fit, impression time (IT), and patient satisfaction through a VAS scale. We decided to evaluate these clinical parameters because of their importance in daily practice, providing practical implications for the clinician.

## 2. Materials and Methods

A randomized controlled trial (RCT) was carried out at the University of Bari “Aldo Moro” and Magna Graecia Institute (Taranto), Italy, in accordance with CONSORT criteria. The study was conducted in accordance with the Declaration of Helsinki, and the protocol was approved by the Local Ethical Committee of Hospital “IRCCS Giovanni Paolo II” (Study n. 1355/CE). The study was registered at ClinicalTrials.gov (NCT06215781).

All participants signed informed consent forms before being included in the study.

The entire sample was a convenience sample of patients who met the inclusion criteria.

### 2.1. Inclusion and Exclusion Criteria

Inclusion criteria were the following:good oral health;no dental caries;no periodontal disease;completed follow-up from 6 months to 1 year after delivery of prosthetic restorations;no parafunction;general healthy status.

Exclusion criteria were the following:bad oral health;presence of dental caries or periodontitis;patients with parafunction (bruxism) and TMJ pathologies (condylar meniscal incoordination, ankylosis, joint click);bad general health status;failure to complete the follow-up period.

### 2.2. Surgical Steps

The first diagnostic assessment required orthopantomography (OPG) and small FOV Cone Beam Computed Tomography (CBCT). OPG and CBCT were performed with the same device, a NewTom GiANO HR (NewTom/Cefla S.C., Imola, Italy). To reduce prosthetic and surgical bias due to incorrect 3D implant placement, guided surgery was performed with DTX Implant Studio (Version 3.2, Nobel Biocare, Zurich, Switzerland) software. Following the delayed placement protocol, the single implant was placed at least 3 months after tooth extraction. Implants were loaded after 3 months from placement to allow their complete osteointegration, evaluated with an intraoral Rx (Sopro Imaging, La Ciotat, France). In particular, the clinicians inserted the following implants: Nobel Parallel TiUltra CC (Nobel Biocare, Zurich, Switzerland), Nobel Active TiUltra CC (Nobel Biocare, Zurich, Switzerland), Nobel Biocare N1 (Nobel Biocare, Zurich, Switzerland), and NobelReplace (Nobel Biocare, Zurich, Switzerland).

### 2.3. Study Design

The patients were randomly allocated to three arms that followed 3 prosthetic workflows:Fully digital workflow (DG);Combined analog–digital workflows (ADG);Fully analog workflow (AG).

The study arms were formed with “simple randomization” whereby each patient was assigned to a group by using a list of random numbers generated by specialized online software (Research Randomizer, by Geoffrey C. Urbaniak and Scott Plous, https://www.randomizer.org (accessed on 23 January 2018)).

#### 2.3.1. Fully Digital Workflow Arm

Digital impressions were performed with Trios 3 (3Shape, Copenhagen, Denmark), following the manufacturer’s instructions, whereby the scanning path provides oscillating movements to record the occlusal and lateral surfaces. Elos Accurate Scan Body Nobel CC (Elos Medtech, Gothenburg, Denmark), Position Locator N1 TCC e Position Locator N1 base (Nobel Biocare, Zurich, Switzerland) were used as scanbodies. All scans were exported in STL format and then used as virtual test images. Final crowns were made of monolithic colored zirconia CAD-CAM milled with Nobel Procera Crown Zirconia (Nobel Biocare, Zurich, Switzerland).

A clinical case of a 2.5 (NobelReplace) with fully digital workflow is reported below (Figure 1). 

#### 2.3.2. Combined Digital–Analog Workflow Arm

The definitive impression was performed with Hydrorise (Zhermack, Badia Polesine, Italy), a hydrophobic polyvinylsiloxane, in putty and light formulation, using the single-phase two-component technique. The analog impressions were cast in type IV GC Fujirock Gold (GC Corporation, Tokyo, Japan) precision plaster. A DOF Swing laboratory scanner (DOF, Inc., Seoul, Republic of Korea) was used to convert the plaster cast into a digital model. A resin provisional was applied when necessary. Final crowns were made of CAD-CAM milled monolithic colored zirconia or cut-back zirconia with ceramic vestibular layering in the esthetically relevant areas.

A clinical case of a 2.1 (NobelReplace) with combined digital–analog workflow is reported below (Figure 2). 

#### 2.3.3. Fully Analog Workflow Arm

The impression was conventional, with an analog material such as hydrophobic polyvinylsiloxane, as in the previous arm. All the models were plaster cast. Definitive crowns provide a metal ceramic crown made by chrome–cobalt alloy substructure with feldspathic ceramic layering. No digital step was performed. A clinical case of a 1.4 (NobelReplace) with fully analog workflow is reported below (Figure 3).

The purpose of this RCT is to compare the clinical efficacy and reliability of these three workflows with each other. Treatment design, implant placement, and prosthetic steps were carried out by the same 3 expert operators. Laboratory procedures were conducted by the same dental technician in order to make the sample as uniform as possible. The clinicians were not blinded but had to collaborate to reach the best result. After the crown delivery, two expert clinicians took clinical measurements, and a satisfaction questionnaire was administered at the end of treatment. These evaluators have decades of experience in the dental prosthetic field, in addition to being academic professors (M.C. and S.C.).

### 2.4. Evaluated Outcomes

Interproximal contact accuracy (IC), occlusal contact accuracy (OC), marginal fit, and impression-taking time (IT) were considered. Moreover, the patient’s perceived comfort during the procedures was assessed by administering a questionnaire with a VAS scale.

#### 2.4.1. Interproximal Contact (IC)

Interproximal contact was defined as the entity of contact between the prosthetic crown and the adjacent tooth, measured by evaluating the resistance to the sliding of waxed dental floss (Essential floss, Oral-B, Procter & Gamble, Cincinnati, OH, USA) in the interproximal space. Since it is not possible to make a mathematical calculation of the force in Newtons (N) we assigned a numerical value between 0 and 4 (where 0 means no contact and 4 means excessive contact such that the floss cannot pass through) to the entity of the interproximal contact considering the resistance opposed to sliding.

Interproximal contact values were as follows:Absent: requires a crown remake;Slight: requires a slight interproximal adjustment;Good: requires no adjustment;Optimal: requires no adjustment;Excessive: requires a slight interproximal adjustment.

#### 2.4.2. Occlusal Contact (OC)

The accuracy of the occlusal contact of the prosthetic crown with its antagonist was assessed using a dichromatic articulating paper (Bausch Articulating Paper, Bausch Inc., Nashua, NH, USA) in order to identify any discrepancies in occlusal contacts, static and dynamic, deserving of correction. For classification purposes, it was necessary to standardize the results into numerical values based on the extent of correction required for the functionalization of the restoration. The clinicians gave a score from 0 to 3, where 0 indicates the absence of contact between crown and antagonists, 1 indicates correct contact between them that needs no correction, 2 indicates slightly altered contact that requires minor correction, and 3 indicates altered contact between the prosthetic crown and its antagonist which requires major correction.

#### 2.4.3. Implant–Abutment Fit (IF)

The presence of a gap between the abutment and the implant screw was assessed through an intraoral Rx performed with a Rinn device with an angle of 0° +/− 10°. This radiological approach is a conventional method of evaluating possible gaps between the prosthetic and implant components, which can reduce the overall prosthetic survival. A score of 0 was assigned if no gap was present, and 1 was assigned if any gap was present.

The intraoral Rx of three patients are reported in Figure 4.

#### 2.4.4. Impression-Taking Time (IT)

The impression-taking time was expressed in seconds (s). For the analog technique, the total calculated time is obtained as the sum of the time for the extraoral mix of the impression material and performing the procedure; for the digital technique, the calculated time consists of the time required to fully scan the arch (time provided by the scanning software, version 21.4).

#### 2.4.5. Comfort

A questionnaire was administered to each patient to evaluate the degree of comfort experienced by the patient during impression-taking procedures. The survey included a VAS scale in order to quantify the degree of comfort felt during the procedure. The authors decided to consider only the comparison between digital and combined workflows to avoid redundancy of results, because the comfort experienced by the patient is critically affected by the type of impression. Combined and fully analog workflows shared the same analog impression procedures; thus, their results are similar.

### 2.5. Statistical Analysis

In order to determine the presence of statistically significant differences between the means of the parameters examined in the groups previously reported, a one-way ANOVA (analysis of variance) statistical model was used. Formally, the null hypothesis is the following: there are no differences between clinical parameters based on different treatment flows. The alternative hypothesis is the following: at least one group presents a significant difference. The error α is set to 0.05. In the case where *p* < α, the null hypothesis H0 is rejected. In cases where the differences between the groups are statistically significant (*p* < 0.05), a post hoc analysis will further be performed using the LSD (least significant difference) Fisher test. This test allows us to assess the presence of a statistically significant difference between each pairwise combination of the groups when LSD values were lower than |µ_1_ − µ_2_| values. Fisher’s contingency test is used to measure parameters with binary values. The data are calculated using the electronic program MATLAB (MathWorks version 23.2, Natick, MA, USA).

## 3. Results

This study lasted 5 years and included 60 patients: 33 females and 27 males. Following the CONSORT criteria, the patients were randomly distributed as shown in Figure 5.

Therefore, a total of 72 dental implants were positioned, and their relative implant-supported crowns were realized. The fully digital group (DG) included 26 crowns, the combined analog–digital group (ADG) included 24 crowns, and the fully analog group (AG) included 22 crowns (Figure 6).

The clinical parameter results are described as follows.

### 3.1. Interproximal Contact (IC)

The range of interproximal contact is between 0 and 4. The evaluation of the interproximal contact of crowns belonging to the different groups showed a significant difference in the ANOVA test (*p* = 0.0003). The results are reported in the following table (Table 1).

After the LSD Fisher test, the means of the intensity of IC of the DG group was found to be statistically higher compared to AG and ADG groups (Table 2).

### 3.2. Occlusal Contact (OC)

The range of OC was between 0 and 3, depending on the intensity. The results of the statistical analysis were similar to the IC values. The ANOVA table shows a significant difference between the three groups (*p* = 0.0009). The OC results are reported in the following table (Table 3).

After the LSD Fisher test, the DG values were shown to be statistically higher than those of the AG and ADG groups (Table 4).

### 3.3. Implant–Abutment Fit (IF)

The implant–abutment fit results showed no significant differences (p=0.5966), as shown in the table below (Table 5).

### 3.4. Impression-Taking Time (IT)

The means of the impression-taking time showed a statistical difference between the three groups (*p* < 0.0001). The results are listed in the table below (Table 6).

The LSD Fisher test demonstrates that the time required to take a digital impression was significantly lower than the time required to take an impression with silicones (Table 7).

### 3.5. Comfort

The Student’s *t*-test reported a significant difference between the patient’s comfort, expressed with a VAS scale, showing better comfort experienced with fully digital procedures than with conventional procedures, as reported below (Table 8).

The results obtained in this RCT are summarized in the following table (Table 9).

## 4. Discussion

This RCT aimed to evaluate the clinical outcomes of three workflows to realize a single implant-supported crown. The null hypothesis was partially rejected: the fully digital, combined analog–digital, and fully analog workflows presented some significant differences.

Full digitalization aims to substitute conventional procedures in every branch of medicine progressively, and the dental field is not excluded. There are many points of strength of “digital conversion”. The in vitro and ex vivo studies are very promising, but the real clinical accuracy of digital techniques is still under debate and understanding must be deepened with further clinical studies [1]. This clinical RCT study evaluates several clinical features after three distinct prosthetic workflows to realize a single implant-supported crown, finding some significant differences. The first outcome that the authors evaluated was the entity of the interproximal contact (IC) of the definitive crown. Interproximal accuracy was analyzed by using waxed dental floss, and the results were standardized into the following four groups: (1) clinically excellent (i.e., floss can pass through the contact points only through the application of strong pressure), (2) clinically acceptable (“slightly strong”) (i.e., less adherent contact, floss can pass through the contact point despite low pressure applied), (3) clinically acceptable (“slightly weak”) (i.e., floss passes through very easily, little resistance is offered), and (4) corrections necessary (i.e., weak contact, a 100 μm metal strip can pass through easily). The importance of a correct IC entity was reported in a comparative study by Naves et al. [14], which confirms that an interproximal contact surface improves the prosthesis biomechanical behavior, leading to better mechanical stress distributions and better patient satisfaction due to less food impaction, resulting in better biological outcomes and improved survival, compared to a contact point with a lesser IC entity. Graf et al. measured the IC with a 8 μm Shimstock foil, finding that a digital cast-free workflow was superior to a conventional one, regarding the time spent and the amount of clinical adjustment [15]. In a similar case series [16], after a fully digital workflow, 3 crowns out of 22 failed interproximal contact, indicating that this type of workflow could be considered reliable.

Few clinical studies have compared the reliability of a digital workflow to that of a conventional workflow. Gao et al. found no IC differences between the two workflows after a modified USPHS criteria [17], according to Delize et al., who considered the same evaluation method of the IC that we used. Conversely, we found a superiority of IC after the fully digital workflow, of which the score was significantly better than the combined analog–digital and fully analog workflows (*p* = 0.0003). The same result was obtained regarding the occlusion contact (OC) between antagonists. The accuracy of OC was measured by articulating paper, standardizing the results into the following three groups: (1) excellent (no need for occlusal adjustments), (2) acceptable (need for minimal adjustments), and (3) adjustments necessary (need for major adjustments). We found that the OC entity of the fully digital workflow was better than the combined and fully conventional ones. OC is usually evaluated with an 8 μm Shimstock strip or marked with 12-μ occlusion foil. We decided to use the latter to better identify the altered contact in centric and eccentric occlusion. The accuracy of the digital workflow regarding the OC entity is still under debate. Some authors found no differences between the two workflows [9,16,18]. In contrast, several authors [19,20] reported that digital workflows presented better occlusal contact than conventional ones at crown delivery, resulting in a saving of time occurring in occlusal adjustment. The better performances of the digital workflow regarding IC and OC is probably due to the role of artificial intelligence (AI), which is pivotal during prosthetic digital planning, and the effectiveness of the digital workflow is confirmed by Lerner et al. in their retrospective study: only 6 crowns out of 106 presented a critical OC alteration, after a fully digital workflow [21]. In this case, AI permits the correction and bypassing of the human skills of dental technicians [11], which are different from each other, depending on the knowledge and experience of the individual, which are necessary to realize a natural prosthesis that mimics the shape and color of a patient’s teeth. IC and OC accuracy have a critical clinical relevance, as they could save or waste time spent in the dental chair, resulting in an economic impact and affecting patient satisfaction. Another variable that definitely affects the time spent in the dental chair is the impression-taking time, defined as the overall time that the clinician needs to take a reliable impression. The comparison between digital and conventional impressions is still being discussed, especially for wide rehabilitation involving FDP or full-arch implants, because of the heterogenicity of the results [22,23]. Still, the latest studies indicate that digital impressions with an intraoral scanner are superior to conventional procedures concerning a single tooth [24]. Moreover, a recent study confirms that digital casts after an intraoral digital impression could be more accurate than conventional gypsum casts fabricated after an elastomeric analog impression of a single implant-supported crown [25]. On the other hand, the accuracy of digital impressions could be affected by several factors, such as the implant position, the type and shape of the scan abutment, the clinician’s skills, and the type of intraoral scanner used [22,26]. The accuracy degree of digital and conventional impressions is especially reflected in the marginal fit of the restoration, which affects their longevity. The clinically acceptable range of crown misfits is up to 200 µm [27], but unfortunately we were not able to measure this gap, as our intraoral Rx do not permit a micrometer measurement due to their sensitivity and image deformation due to an eventual incorrect X-ray tube positioning. To overcome this problem, we decided to consider the misfit as a dichotomic variable: even a slight misfit entity was considered as the presence of a gap. The results of our investigation showed no significant differences between the digital and conventional workflows, reflecting the clinical reliability of digital procedures. This result is consistent with many authors who consider the digital workflow as a viable, faster, and more comfortable alternative to conventional procedures, keeping the same or improved accuracy, when performed by clinicians expert in the field [7,28,29,30]. Besides the technical aspects evaluated in this study, we decided to compare the time required for the digital and conventional impression procedures and the relative comfort perceived by the patient. The reduction in chairside time is an important factor to consider in dental office economics, as it reduces the cost per hour and improves the patient’s perception and satisfaction [31]. Many authors considered the impression time in their comparison between digital and conventional workflows. Park et al. [32], in their clinical study, divided the sample into two groups: in the first group, impressions were taken through conventional methods, and in the second group, they were taken through intra-oral scanners (AEGIS.PO, Digital Dentistry Solution, Seoul, Korea) and CEREC Omnicam (Sirona, Bensheim, Germany). The time required for the digital impressions, respectively, AEGIS (7:16 ± 1:50 min:s) and CEREC (7:29 ± 2:03 min:s), was significantly less than the conventional methods (12:41 ± 1:16 min:s).

In an RCT conducted by Gjelvold et al. [7], 42 patients were divided into two groups based on the impression technique. Among the parameters analyzed, the impression-taking time showed significantly better results for digital scanning (*p* < 0.0001), with a duration of 7:33 ± 3.37 and 11:33 ± 1.56 min for the digital and conventional techniques, respectively.

In the study conducted by Yuzbasioglu et al. [33], 24 students with no previous experience in impression-taking underwent conventional (polyether) and digital (CEREC Omnicam, Sirona) techniques. The objective was to assess the time required for the entire procedure, confirming that the digital workflow presents significantly better results than the analog workflow (248.48 ± 23.48 s digital workflow vs. 605.38 ± 23.66 s conventional workflow).

In contrast, in the RCT conducted by Sailer et al. [34], the overall impression-taking time for a full arch was shown to be reduced in the analog stream compared to the digital method (Lava 1091 ± 523 s, iTero 1313 ± 418 s, CEREC 1702 ± 558 s, conventional 658 ± 181 s). The difference with the previously mentioned studies could lie in the larger area of the impression, which may reduce digital impression accuracy. In a 2022 study conducted by Carneiro et al. [35], 17 participants underwent the placement of 3-4 implants in order to evaluate the time-effectiveness of different impression-taking techniques: two digital methods (scan with scanbodies and scan with scanbody-associated devices) and two conventional methods (solid index or open-tray), reporting a significant reduction in impression-taking time with digital techniques (*p* < 0.0001). Consistent with the previous studies, our study demonstrates a better performance of the digital impression performed by an intraoral scanner (Trios 3, 3Shape), compared to the conventional impression with silicones (*p* < 0.0001). The overall means of digital impression-taking time was 101.58 s, while the analog impression-taking time was 361.71 for the combined digital–analog and 363 for the fully analog workflow. The reduction in time spent in the dental chair and the absence of analog impression materials could influence the comfort perceived by patients during the treatments. Many studies have deepened this aspect: a recent systematic review concluded that digital impressions are faster than conventional ones, improving the patients’ dental experience and comfort [31]. These conclusions are shared between authors, including regarding digital procedures in children [33,36,37,38]. Consistent with them, our results showed that the patients subjected to digital impressions reported a better experience and better comfort than the conventional impression techniques, with a significant *p*-value (*p* < 0.0001), confirming better acceptance of the digital workflow compared to the conventional one. However, this study presents some limitations. The main limitation is represented by the lack of sufficient data in the literature: the parameters used to evaluate the clinical outcomes are not standardized, which makes it difficult to corroborate the results reported in our study. In addition, the lack of other similar clinical RCTs does not allow an adequate comparison. Moreover, the sample was a convenience sample, resulting in a small number of evaluated patients. The clinicians were not blinded and the heterogenicity of procedures, especially in the fully analog arm, depends on clinician and dental technician skills, affecting the overall outcomes, which, however, is present in daily clinical practice. Finally, this study lacks long-term follow-up, because it finished in 2023; however, we want to present our preliminary results due to our innovative findings, which may be followed by a further study with long-term follow-up.

Further clinical studies with a longer follow-up may move in this direction, considering all the difficulties that investigators must overcome in studies that involve patients.

## 5. Conclusions

The null hypothesis of this study was rejected: fully digital, combined analog–digital and fully analog workflows presented some significative differences. The fully digital workflow presented significantly better results for interproximal contact, occlusal contact, impression-taking time, and comfort perceived by patients, compared to the combined analog–digital workflow and fully analog workflow, which presented no difference between each other. The implant–abutment fit presented no differences between groups. The reported results encourage the use of all-digital workflows in dental practices, allowing a revolution in the field, improving diagnosis, treatment, and patient experience. Digital workflows could be considered a viable and reliable alternative to common analog prosthetic procedures when performed by clinicians expert in digital dentistry.

## Figures and Tables

**Figure 1 jfb-15-00149-f001:**
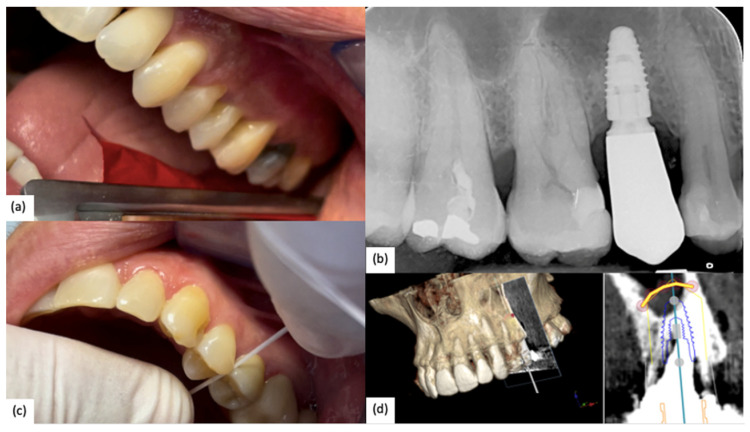
(**a**) Occlusal contact check; (**b**) Intraoral Rx confirms a correct implant–abutment fit; (**c**) Interproximal contact check; (**d**) Digital planning of implant insertion. The yellow line represents the expected maxillary sinus lift.

**Figure 2 jfb-15-00149-f002:**
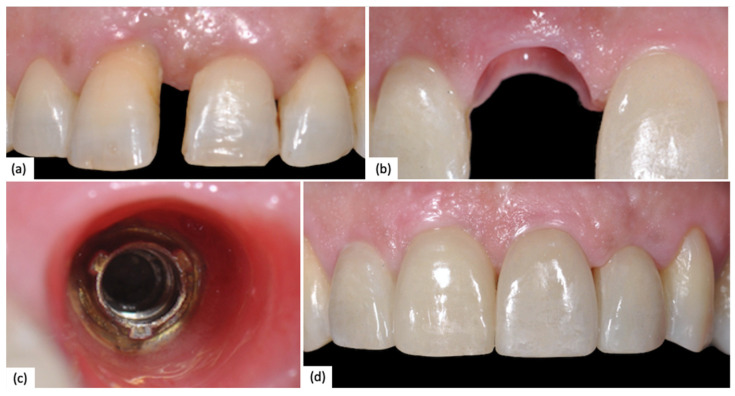
(**a**) Preoperative clinical situation; (**b**) Peri-implant tissue after provisional conditioning; (**c**) Peri-implant tissue before polyvinylsiloxane impression; (**d**) Final restoration.

**Figure 3 jfb-15-00149-f003:**
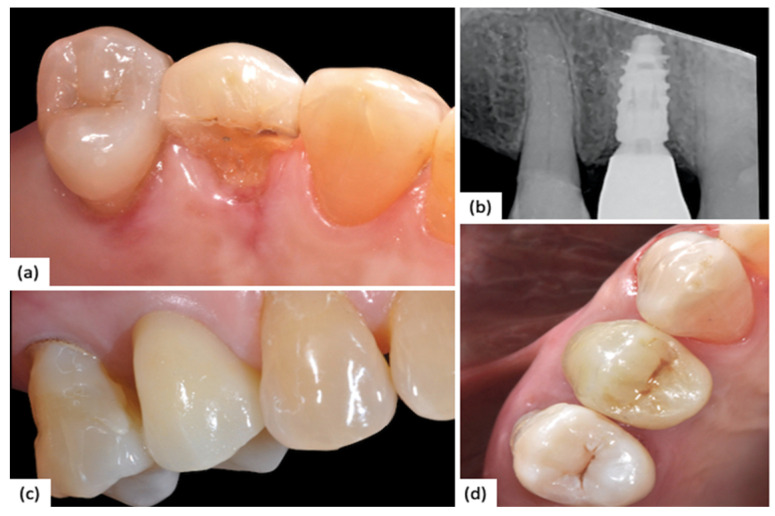
(**a**) Preoperative clinical situation showing a vertical fracture of 1.4; (**b**) Intraoral Rx showing no marginal implant–abutment gap; (**c**) Vestibular view of final restoration; (**d**) Occlusal view of final restoration.

**Figure 4 jfb-15-00149-f004:**
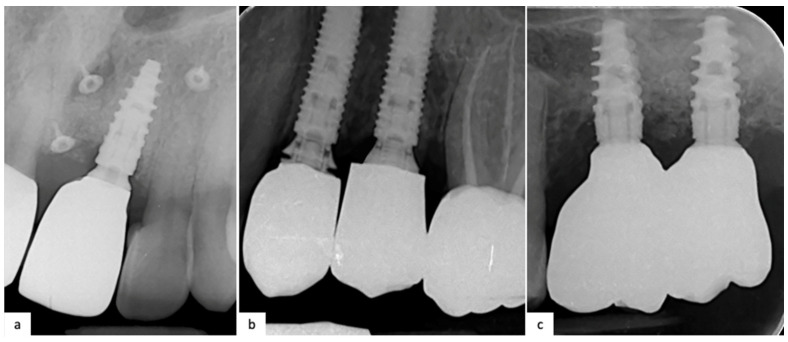
Intraoral Rx: (**a**) 2.1 after 1 year which followed the fully digital workflow after a GBR (Nobel Biocare N1); (**b**) 2.5 and 2.4 after conventional workflow that shows an implant–abutment misfit of 2.4 that needs to be corrected (Nobel Parallel TiUltra CC); (**c**) 1-year follow-up Rx of 2.7 and 2.6 which followed a combined analog–digital workflow (Nobel Active TiUltra CC).

**Figure 5 jfb-15-00149-f005:**
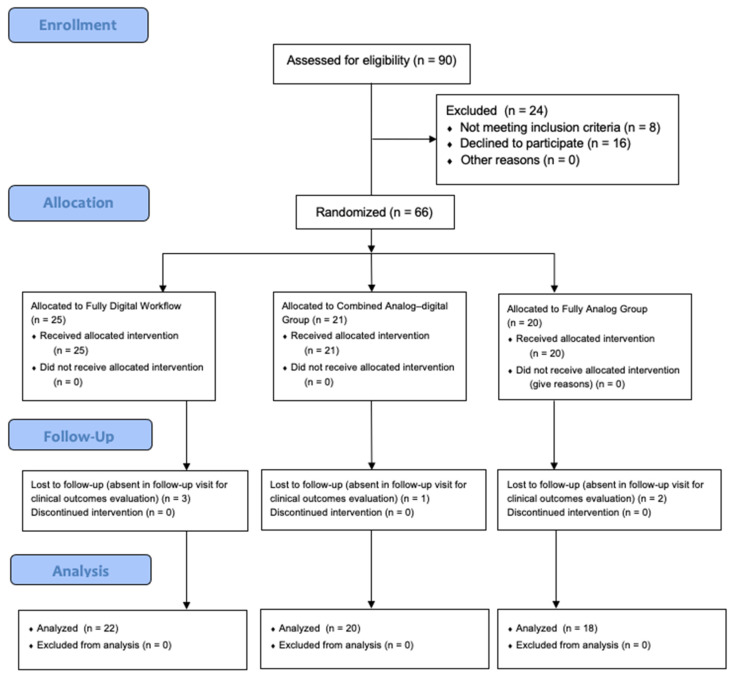
CONSORT flowchart.

**Figure 6 jfb-15-00149-f006:**
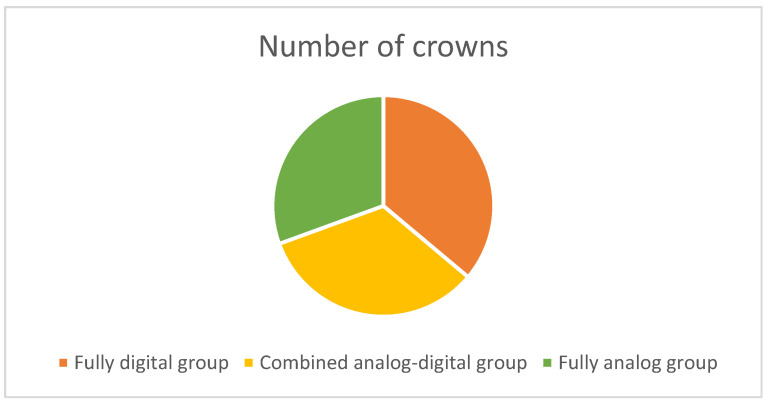
Crown distribution.

**Table 1 jfb-15-00149-t001:** Interproximal contact.

Groups	N.	Overall Means	Standard Deviation	Standard Error
Fully digital	26	2.9231	0.891	0.1747
Combined analog–digital	24	2.0833	0.6539	0.1335
Fully analog	22	1.9545	0.9989	0.213

**Table 2 jfb-15-00149-t002:** LSD Fisher test between groups.

Groups	|µ_1_ − µ_2_|	LSD	Significance
Fully digital vs. combined	0.8477	0.4838	Significant
Analog vs. combined	0.1288	0.5044	Not Significant
Analog vs. digital	0.9686	0.4951	Significant

**Table 3 jfb-15-00149-t003:** Occlusal contact.

Groups	N.	Overall Means	Standard Deviation	Standard Error
Fully digital	26	1.3077	0.6177	0.1211
Combined analog–digital	24	2	0.417	0.0851
Fully analog	22	1.8636	0.8888	0.1895

**Table 4 jfb-15-00149-t004:** LSD Fisher test between groups.

Groups	|µ_1_ − µ_2_|	LSD	Significance
Fully digital vs. combined	0.6923	0.3731	Significant
Analog vs. combined	0.1364	0.3891	Not Significant
Analog vs. digital	0.5559	0.3819	Significant

**Table 5 jfb-15-00149-t005:** Fisher exact test.

Groups	Gap Absence	Gap Presence
Fully digital	21	5
Combined analog–digital	17	7
Fully analog	15	7

**Table 6 jfb-15-00149-t006:** Impression-taking time means.

Groups	N.	Overall Means (s)	Standard Deviation	Standard Error
Fully digital	26	101.5769	14.4227	2.8285
Combined analog–digital	24	361.7083	12.8857	2.6303
Fully analog	22	363	11.5635	2.4653

**Table 7 jfb-15-00149-t007:** LSD Fisher test.

Groups	|µ_1_ − µ_2_|	LSD	Significance
Fully digital vs. combined	260.13	7.3936	Significant
Analog vs. combined	1.2917	7.7094	Not Significant
Analog vs. digital	261.42	7.5663	Significant

**Table 8 jfb-15-00149-t008:** Student’s *t*-test.

Groups	Means	Standard Error	*p*-Value
Fully digital	1.48	1.40	<0.0001
Combined analog–digital and fully analog	4.23	1.84	

**Table 9 jfb-15-00149-t009:** Overall results.

Outcomes	Fully Digital vs. Combined	Analog vs. Combined	Analog vs. Digital
Interproximal contact	Significant	Not Significant	Significant
Occlusal contact	Significant	Not Significant	Significant
Implant–abutment fit	Not Significant	Not Significant	Not Significant
Impression-taking time	Significant	Not Significant	Significant
Comfort	Significant	Not Significant	Significant

## Data Availability

Data are unavailable due to privacy restrictions.

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
