# Peer review of "Comparison between Conventional and Digital Workflow in Implant Prosthetic Rehabilitation: A Randomized Controlled Trial"

_jfb, 2024, doi:10.3390/jfb15060149_

Round 1

Reviewer 1 Report

Comments and Suggestions for Authors

In current manuscript, authors aimed to perform a two-arm RCT between the entire digital, the combined digital-analogic, and the entire analogic workflows of the implant-supported prosthesis through the evaluation of restoration outcomes using a VAS scale. The idea is novel and manuscript is well organized. However, there are a few small points that need attention:

1.     The full form of RCT should be mentioned in the Introduction

2.    Figure1 is not clear

3.     There needs to be an explanation regarding why the comfort comparison is made between the fully digital and combined digital-analog groups and not among three various groups.

4.     The methodology for determining the sample size and whether the sample size is sufficient needs to be discussed.

Author Response

Dear Reviewer #1

Thank you very much for your valuable comments. You can find in the attached document our point-by-point replies. 

Best Regards

Reviewer 2 Report

Comments and Suggestions for Authors

Dear Editors and Authors, Thank you for the opportunity to review this interesting study.

The authors conducted a Randomized Controlled Trial (RCT) study that includes and compares three arms: the full digital, combined digital and analogue, and full analogue dental implant workflow. The study is well-conducted and well-written. However, for publication, I have the following comments:

1-Abstract: In line 17, you stated that this is a three-arm RCT. However, in the introduction, line 63, you wrote it as a two-arm RCT. Please revise this to avoid any confusion or misunderstanding.

2-Section 2.1 (Inclusion Criteria): You included the exclusion criteria under the inclusion criteria without a separate subtitle. Please revise.

3- For every hardware or software mentioned (e.g., OPG, CBCT), please include the company name, city, country, and version used.

4-Clinical Cases: I recommend inserting clinical cases under each arm in sections 2.3.1 to 2.3.3. Providing three cases for each therapy arm would enrich the study and offer practical examples of the workflows.

5- Please provide information about the evaluators, including whether there was one or more and details about their calibration.

Comments on the Quality of English Language

The english language is fine.

Author Response

Dear Reviewer #2,

Thank you very much for your appreciation words. You can find the authors point-by-point replies in the attached document. 

Thank you

Best regards
